# Understanding Green Innovation: A Conceptual Framework

Jacob Guinot * , Zina Barghouti  and Ricardo Chiva 

Department of Business Administration and Marketing, University Jaume I, 1071 Castelló de la Plana, Spain; barghout@uji.es (Z.B.); rchiva@uji.es (R.C.)
* Correspondence: guinotj@uji.es

**Abstract:** In recent years, the growing emergence of environmental problems has meant that sustainability and related concepts such as green innovation have acquired special importance. This has resulted in a significant body of literature addressing these concepts. To help to integrate this extensive literature and establish a theoretical framework, this study summarizes the main principles and roots of green innovation. To this end, this study first makes a generic theoretical approach to the concept of innovation. Then, due to its direct link with green innovation, emphasis is placed on the importance of the value of sustainability in companies. After that, the meaning and current relevance of green innovation in today's business environment is addressed. Finally, the main precepts and fundamentals of green innovation are established, and a series of academic proposals are made to further advance the study of this concept. This theoretical review may serve as encouragement to further research the concept of green innovation and contribute to providing a clarifying and comprehensive view of this topic.

**Keywords:** green innovation; innovation; sustainability; environmental

## 1. Introduction

The environmental impact of human activities has become a global and growing concern for society. In recent years, carbon dioxide emissions from fossil fuels have reached record highs, global average sea levels have risen by 20 cm from 1900 to 2018, and heatwaves, forest fires, and air pollution have increased. In addition, concentrations of major greenhouse gases have continued to increase in the last few years and the global mean surface temperature for the period 2017–2021 is among the warmest ever recorded. Furthermore, the frequency and intensity of extreme weather and climate events have increased in all regions of the world. This has led to a growing need for governments to implement measures to reduce environmental damage [1,2]. However, solving environmental problems is not just a matter of government policy, organizations also play a very important role in reducing these problems [3,4]. In fact, in the current business market, organizations must respond effectively to a double trend [5]. On the one hand, organizations should achieve efficiency and market share through constant innovation, which requires resources, capabilities, and investments. On the other hand, organizations should achieve a certain degree of ethical responsibility and coherence with the society in which they operate. Therefore, business today is a question of finding a balance between competitive adjustment (innovation) and legitimacy adjustment (sustainability and respect for the environment).

Several researchers (e.g., [5–7]) have proposed a type of innovation that achieves this balance and reduces environmental damage while increasing business efficiency. This is the idea that encompasses the proposed concept of green innovation (GI). GI is understood as the search, promotion, and development of ecological products, services, and processes that incorporate a certain degree of novelty [1]. It consists of new innovative ideas that aim to provide a product or service without attacking nature, respecting the environment, and being ecological in its conception, construction, use, and disposal [8].

GI is an issue of great relevance for organizations as consumers are increasingly demanding measures from companies and institutions to reconcile this balance between production and environmental sustainability [6,9,10]. In fact, the literature indicates that, at present, GI is considered a business strategy to obtain a competitive advantage since it contributes to opening new markets or developing new products and applications sustainably and responsibly [5,7]. The importance of this type of innovation lies in the fact that it allows companies to compete in the market using clean and efficient technologies and facilitates adaptation to new sustainable patterns of consumption. As pressure to comply with environmental regulations and meet consumer environmental preferences has increased in the last few years, GI has been turned into a source of competitive advantage [7].

As a result, the concept of GI has attracted particular interest in academia. This interest has resulted in a growing body of literature that has attempted to gain an indepth understanding of the main implications of the concept. Because of this explosion of academic work, we are faced with literature that is certainly fragmented and poorly integrated in terms of its conceptual framework. Thus, to consolidate a theoretical framework that integrates existing literature, this study offers a complete review of the concept of GI. This theoretical review will make it possible to underpin the main precepts, components, and proposals of GI. Thus, researchers, and all those interested in understanding and studying GI, will find in this study a starting point for acquiring the theoretical foundations of the concept and adopting a conceptual framework of reference. Despite the progress achieved in this field of research and the growing attention that this subject has received in recent years, there are very few literature reviews on the concept of GI. The conceptual review carried out in this study, linking GI with its foundations, as well as shedding light on its forms of measurement and application in public institutions and technology centers, is a pioneering work.

This study is broken down into the following parts: First, the concept of innovation and its main variants or forms are introduced. Second, given its close linkage with the concept of GI, the concept of sustainability and its current importance in business management is addressed. Next, the subject of GI is analyzed indepth, including a conceptual review, the role of management in promoting GI, ways of measuring GI, and the role of technology centers in the GI processes. Finally, a discussion section summarizes the main conceptual bases and theoretical foundations provided in the work and sets out the main challenges that exist in this field of study, including a series of proposals for future lines of research.

## 2. Concept and Forms of Innovation

### 2.1. The Concept of Innovation

Innovation has long been considered an essential tool for organizational competitiveness and success and there is a great amount of literature on the topic (e.g., [11–13]). The Oslo Manual [14] defines innovation as the conception of, or significant changes in, the product, process, marketing, and organization of the company to improve results and performance. Innovation is considered a complex and creative process, which consists of the creation of a new product, process, or the introduction of a new method of organizational management [15]. Authors such as Kahn [16] indicate that innovation is understood as a process that involves the development and application of creativity, aimed at satisfying consumer tastes and needs, through something new or improved. In similar terms, Varadarajan [17] suggests that innovation is like a mutation that breaks the monotony of economic cycles and drives continuous growth, and that goes hand in hand with technological development.

According to Henderson [18], innovation is a phenomenon that makes it easier for companies to penetrate markets and conquer new markets faster. However, innovation does not happen automatically but requires a mixture of vision, passion, enthusiasm, will, energy, and constant hard work that transforms good ideas into reality [19]. Additionally, being involved in innovative actions also contributes to individual growth since innovation requires being proactive and confident in taking risks and making decisions [20]. Therefore,

innovation does not only influence the creation of new products, processes, etc., but also improves and develops the capabilities of individuals.

The literature distinguishes two main types of innovation: radical innovation and incremental innovation. Incremental innovation refers to the creation of value on an existing product or process by incorporating new improvements [21,22]. Incremental innovation is characterized by gradual changes in the product, service, or process, and achieving results in short periods [23]. This innovation arises when the market demands specific characteristics of an existing product or other production requirements. This type of innovation in business is a common strategic practice, allowing them to continue to survive and increase their profits [22,24]. In contrast, radical or disruptive innovation refers to the creation of a new product or service not previously known, i.e., creating something that is not yet on the market [21,25]. This kind of innovation requires high investments, dedication, time, research, and development [22]. For this reason, companies often choose incremental innovation rather than radical innovation. Table 1 summarizes the basic conceptual differences between these two types of innovation.

**Table 1.** Differentiation between radical and incremental innovation.

| Radical Innovation | Incremental Innovation |
| --- | --- |
| Proposal of a new product or service | Proposal of an improvement of an existing product or service |
| High level of risk and uncertainty | Low level of risk and uncertainty |
| Application of new technologies | Minimal change to existing technologies |
| Can generate drastic changes in a market, industry, or company | Easily adaptable to an existing market |

### 2.2. Forms of Innovation

Several studies and authors discuss the forms of innovation, but the most relevant theory in the field of business is the Oslo Manual developed by the Organization for Economic Cooperation and Development (OECD) in 2005. This manual distinguishes four forms of innovation: product innovation, process innovation, marketing innovation, and organizational innovation. This classification applies to both industry and services, including public services.

Firstly, product innovation is understood as the creation of new products or services or the improvement of existing products and services [26]. The Oslo manual specifies that product innovation occurs when appreciable changes are made to the design, resulting in improvements in the particularities of the operation or use for which the product was created. Product innovation is achieved with knowledge or technology, which are used to apply improvements in materials and components [27]. However, the existence of some changes to the product could be also considered as a marketing innovation.

Secondly, marketing innovation consists of applying marketing methods not previously used by the organization that entails relevant changes in design, packaging, positioning, promotion, etc. [11]. This form of innovation is mainly focused on increasing sales [28]. The Oslo Manual [29] indicates that this type of innovation implies opening new markets, and it also requires changes in the 4Ps (product, price, distribution, and communication). According to the Oslo Manual, for the 4Ps to be considered innovation, the product must have a significant change in the design and packaging; in the price, where it must have an equally significant change based on demand and the options offered; in the distribution, when, for example, new sales channels are implemented, such as the implementation of franchises; in the promotion, when a change is made to the logo, or when CRM (Customer Relationship Management) is implemented.

Thirdly, process innovation is achieved through major changes in techniques or software, to reduce unit costs of production or distribution and enhance the quality, production, or distribution of new or improved products [30,31]. For example, improvements in information and communication technologies (ICT) are considered a process innovation because

they improve the efficiency and quality of a supporting activity. Therefore, the main objective of process innovation is to reduce unit production costs and increase productivity [30].

Fourthly, organizational innovation consists of significant changes in the management, practices, and procedures of the company [32]. It entails changes in the division of labor, strategy, or structure [33]. Furthermore, the updating of knowledge management belongs to this type of innovation, as does the introduction of management systems for production, supply, and quality management operations [34]. Likewise, improvements in customer and supplier relations are considered an organizational innovation [33].

It is interesting to identify and conceptualize these various forms of innovation to understand how companies can be eco-innovators using different organizational decisions, practices, and procedures. Thus, GI can involve all four forms of innovation, depending on the approach to its implementation. In addition, as explained above, a key component of GI is its contribution to sustainable development as it seeks economic development that achieves an environmental balance. Specifically, GI is directly related to the environmental dimension of sustainability, as will be addressed in the following section.

## 3. Sustainable Development: Conceptual Framework and Importance

### 3.1. What Is Meant by Sustainability?

The terms sustainability and sustainable development are used very frequently nowadays and are intrinsically linked to the business sector. Sustainable development is a concept defined in the 1987 Brundtland Report, which refers to "development that meets the needs of the present generation without compromising the ability of future generations to meet their own needs". In other words, sustainable development is understood as meeting the needs of today, without compromising the ability of future generations to meet their own needs [35]. Sustainable development is a phenomenon that improves the quality of life and the natural environment, thus progressing without destroying the livelihood of future generations [36,37].

In the search for definitions of the concept of sustainability, we find the author Elkington [38] who suggests that the concept of sustainability is a difficult task to achieve, and requires some basic principles which are human rights, equity, justice, diversity, rights of future generations, democracy, citizen participation, and economic vitality, among others. Achieving these principles requires the collaboration of institutions, companies, and citizens. In other words, promoting sustainability or sustainable development is not only a matter of the policies established by the government but is the collaboration of all parties, including businesses and citizens [39]. Sustainability also requires three dimensions that must be in balance: economic sustainability, environmental sustainability, and social sustainability [40,41] (see Figure 1). Economic sustainability refers to the use of practices and activities that are economically profitable and also socially and environmentally responsible [42,43]. This means seeking growth and economic profitability without forgetting social welfare and environmental care. Environmental sustainability refers to the efficient and rational use of natural resources to improve current social welfare without compromising the quality of life of future generations [40,44]. Finally, social sustainability refers to the achievement of a balance between the two dimensions mentioned above without generating poverty, exclusion, and inequality [45]. Hence, social sustainability involves the achievement of equity and social justice by promoting the participation of societies in the generation of wealth [46].

Therefore, the term sustainability is understood as the search for a balance between the economic, social, and environmental dimensions [47]. When sustainability is applied in business, it is aimed at finding a balance between the creation of wealth and the use of different human, material, natural, and economic resources [48]. Thus, business sustainability aims to improve the socio-economic conditions of all parties and care for the environment [49]. For this reason, an organization is said to be sustainable when it has the capacity to ensure its continuity and long-term positioning, as well as cooperate in the progress of the present and future generations [50]. Sustainable companies are those

that focus on the development of a formula for profitability from a balanced approach, by creating responsible linkages with all stakeholders and the natural environment [51,52]. These are therefore companies that are characterized by multiple orientations and complete commitment (environmental, social, administrative, and financial).

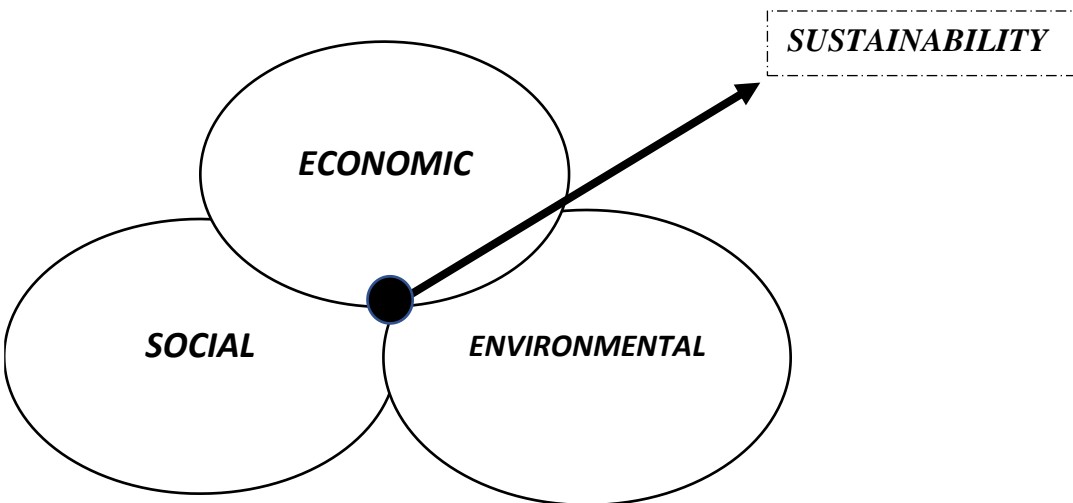

**Figure 1.** Dimensions of sustainability.

Sustainability can be considered one of the great challenges facing humanity today. As we have mentioned, addressing the concept of sustainability includes not only engaging with environmental issues but also making decisions to improve social and economic issues in society. The latter includes aspects such as ending poverty and hunger, improving the health and well-being of citizens, promoting quality education, achieving gender equality, reducing inequalities, promoting decent work and economic growth, and pursuing peace and social justice. These goals are precisely those advocated by the 2030 Agenda for Sustainable Development designed by the UN and constitute a call to urgently address the problem of sustainability. This transition towards sustainability requires a global commitment, in which governments commit themselves to ensure a leadership dedicated to sustainability and provide the necessary resources to undertake actions in this regard. It also requires a commitment at the local level where cities, governments, and local authorities move to achieve this transition with policies, budgets, and regulations that facilitate it. Finally, the commitment of individuals and organizations is necessary to promote sustainability. This requires that professionals, companies, academics, civil society, the media, trade unions, and other stakeholders engage in the necessary actions to contribute to sustainability.

*3.2. The Importance of Sustainability in Organizations*

In recent decades, the capitalist economic system has promoted mass consumption that exalts and embraces the ideals and values of the fleeting, carefree, and hedonistic [53,54]. The capitalist system is based on seeking unlimited growth and is oriented to continuous renewal, cost outsourcing, and the constant generation of new needs among consumers [55]. It is built on a chain of production and consumption that follows a linear system [56]. Its essence is "produce, sell and throw away" [57]. So, materials are extracted, transformed, distributed, and finally sold to become short life cycle "use and waste" products. It is a system that has enriched many companies and nations but has also led to a very worrying environmental imbalance [58,59].

In the last few years, an increasing number of companies and consumers have realized the need to change this unsustainable model [60,61]. This has led to a greater social and environmental sensitivity among companies which is expressed in their management policies and business philosophy. Sustainable companies are motivated by achieving zero environmental impact through designing products and services that have zero net impact.

Therefore, their actions are not only aimed at compensating for the damage caused by economic activity, but also a form of social responsibility aimed at improving the state of the natural environment [51]. This transition to a more sustainable business model involves the transformation from a linear and degenerative system to a circular and regenerative one [62]. It consists of applying a model that adopts the pattern of nature itself. The aim is to imitate the cyclical and reuse processes that set the ecological standard [63]. It entails a regenerative economy design based on formulas for reuse, repair, reconditioning, and ultimately recycling [57].

While traditionally the fundamental objective of companies had been to remunerate shareholder investments, the objective of sustainable organizations is to provide value to all parties involved or affected by the operation of the company (stakeholders) [64]. This opens a new corporate era in which companies must generate value not only for shareholders, but also for customers, workers, suppliers, the community, and the environment around them (shareholders). This change means that social responsibility would become a defining element of business purpose. In this sense, a sustainable organization should meet the needs of the present without compromising the future [65].

In turn, an increasing number of customers, investors, workers, and citizens are demanding a social commitment from companies [9,10]. Because of this, a company's values and principles are increasingly considered key elements in differentiating companies [66]. Hence, companies can be viewed as attractive due to their high social and environmental awareness triggered by business strategies. These business strategies are expressed through the creation of sustainable products and services, based on a genuine commitment to all stakeholders and the environment [9], and not simply on philanthropic actions. Thus, it encompasses companies that are consistent with a set of sustainable and ethical values and do not apply responsibility policies only to save their reputation. In this new economic paradigm, companies must assume environmental responsibility for their actions and find solutions to minimize the negative impact of their actions on the environment [65]. These companies are integrated into the surrounding community and are concerned with generating a better world for all [67].

Sustainable models are being implemented in cases of businesses such as those associated with the economy of the common welfare, cooperatives, those linked to fair trade, or others belonging to the circular economy movement [68]. The success of some of these businesses reveals that it is possible to move away from the neo-liberal economic model and its hyper-consumerist values and go towards an economy of the common good based on the value of environmental sustainability, solidarity, and social justice. These companies that share the importance of sustainability as one of their core values have GI as a primary tool. Hence, GI has become a common strategy for companies oriented toward the common good and concerned with sustainability issues.

## 4. Green Innovation: Review of the Concept, Role of the Governing Bodies, and Measurement

### 4.1. The Concept of Green Innovation

GI has become one of the most important strategic tools for effective sustainable development [69]. In the past, investing in environmental activities was unnecessary, however, strict environmental regulations and widespread environmentalism have changed the competitive rules in practice [1]. This has led to an expansion of methods and processes linked to environmental sustainability, such as GI. Moreover, given that in recent years consumer awareness of sustainability has been growing, many companies have used the green agenda as an engine for achieving a business model based on GI [70]. Thus, GI practices are a logical response to the requirements of the customer, who is willing to pay more for sustainable products and who expects increasingly responsible products and services from firms [1,71]. In fact, previous research (e.g., [6,72]) indicates that GI initiatives contribute to the improvement of a company's performance and competitiveness

as GI practices increase product value and create competitive advantages using ecological differentiation.

As noted above, the literature suggests that innovation is the sum of "design + product". Accordingly, GI can be defined as the sum of "eco-design + eco-production" as GI involves the development of ecological products or processes, applying innovations in technologies that involve energy saving, pollution prevention, and ecological product designs [1,72]. Similarly, Wang et al. [26] define GI as those innovations that focus on achieving sustainable development and the conservation of natural resources through the development of greener products and services. Along the same lines, some authors (e.g., [73,74]) define green innovation as the development of sustainable products and processes through the use or adoption of environmentally friendly raw materials during the manufacturing or design process. This process also involves the application of the principle of eco-design or eco-production. That is to say, the environment is taken into consideration from the moment a product or process idea is conceived, and not only at the end of its useful life [75]. Accordingly, although GI has mainly focused on production processes some companies have expanded this trend not only by redesigning their production but all of their processes, including distribution channels and after-sale service [56]. Hence, GI is an innovation that is included in the entire business cycle, that is, in the design, production, supply, and end-use of commercial products, which mainly contribute to environmental sustainability [76,77].

It can therefore be noted that the main objective of GI is to reduce human-induced material flows and promote sustainability goals [78]. GI includes the adoption of "eco-innovation" in corporate practices, which means the implementation and development of innovations that minimize environmental and social damage and produce economic improvements. This eco-innovation stemming from GI is understood as a balance between the different pillars of sustainability (see Figure 2).

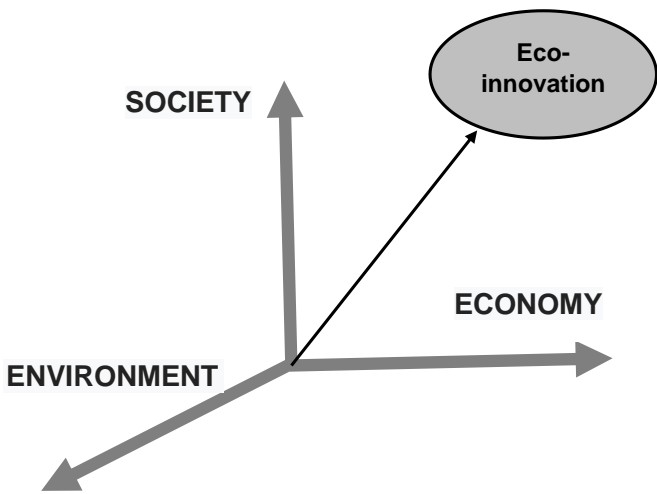

**Figure 2.** Eco-innovation position in sustainable development.

Furthermore, the use of environmentally friendly raw materials, or what is known as "eco-friendly using", which goes hand in hand with GI, leads to several relevant benefits that can differentiate the company from competitors [79]. These benefits include cost reduction, improved company reputation, increased business opportunity, and the attracting of new investors and customers [79,80]. Therefore, GI is associated with better environmental management and environmental performance. In this way, GI in processes and products not only reduces the negative environmental impact of the business but also increases organizational performance through waste and cost reduction [69]. A paradigmatic case of a company that applies the fundamentals of GI is Patagonia Inc. This company manufactures sportswear following a management philosophy based on reducing negative

environmental impact by creating recyclable, reusable, and durable products. This requires that they apply GI in the design of their products so that they can achieve an environmentally sustainable product. This would be a good example of a company that follows the precepts of what is meant by environmental management and has been commonly held up as a model of companies that apply eco-innovation [81].

### 4.2. The Role of Governing Bodies in Promoting Green Innovation

As has been indicated above, to promote competitiveness based on sustainable production, it is crucial that technology and innovation policies in the public and private sectors are aligned. For this purpose, governing bodies can intervene in GI policy to mitigate the "blocking" effects [82] of the closed cycle that limit the implementation of disruptive eco-innovative solutions and make their use more difficult, as new technologies may not be easily compatible with current ones. Moreover, in 2004, to advance the implementation of GI, the "Environmental Technology Action Plan (ETAP)" placed GI among the priorities of European technology policies supported by the European Lead Market Initiative (LMI) and created the Eco-Innovation Observatory (EIO). In turn, the community guidelines on state aid for environmental protection provided for the possibility of aid in the case of GI, and the European Commission facilitated the dissemination of good practices in this field and published several documents and reports. However, the volume of funding to support innovation has been modest and there are gaps in the current funding and promotion programs, such as the Competitiveness and Innovation Framework Programme (CIP), the Seventh Framework Programme (FP7), and the Environmental Technologies Action Plan (ETAP).

From a regulatory point of view, at the European level in GI, we must mention the Directive of 21 October 2009, which establishes a framework for setting eco-design requirements for products related to energy consumption and electrical appliances. Despite the limited amount of funds allocated to GI, the European Union is one of the most important players in supporting the eco-industry and the green sector. In addition, the "Eco-Innovation" program is the latest instrument of an encouraging policy aimed at strengthening Europe's environmental and competitive position by supporting innovative solutions that protect the environment and create a wider market for "green" technologies, management methods, products, and services. By focusing on institutions at the national level, for example, in Spain the reference framework is the "State Innovation Strategy", the cooperation between the public and private sectors and the coordination of actions in different regions as a priority is established.

Gibbs et al. [83] suggest that governing bodies have the mission of defining priorities, regulating, monitoring, and promoting the interest of the private sector towards long-term decision making and the implementation of eco-innovative processes. Therefore, governing bodies play a very important role in the promotion and creation of organizations that develop GI. Many companies are still unaware of the benefits and importance of GI, especially small businesses, or SMEs, and those that know the importance of GI find it difficult to finance the necessary investments. Therefore, despite the creation of these laws and government support, there is still a need for a greater institutional drive to support this type of innovation. More institutional economic support is particularly necessary to overcome existing financing barriers and create organizations with a culture based on GI or eco-innovation.

### 4.3. Green Innovation Measurement

GI can be measured like any other type of innovation. However, its essential component of eco-efficiency makes it difficult to evaluate actions in their entirety because it requires the measurement of different innovation factors (social, economic, and environmental) throughout the innovation process, both radical and incremental [1]. Specific indicators are therefore needed to measure GI, and several international institutions have chosen to use indicators or lead systems to measure eco-innovation, which are mentioned

below (Table 2). It should be noted that these indicators are the most widely used, although different ones can be found.

**Table 2.** GI indicators used by different institutions.

| Indicators | Institution | Applicability |
| --- | --- | --- |
| Group of indicators | OCDE | Environment |
| Innobarometer | European Commission | State of innovation |
| Sustainable Development Indicators Group (SDI) | Eurostat | Development R + D + i, Environment |
| Living planet Index and Ecological Footprint | WWF | Environment |
| Corporate Social Responsibility | Global Reporting Initiative | Organizations |
| ISO 26000 Standardization Standards | International standardized organization | Organizations and public entities |
| Group of indicators of the objectives of Millennium Development | United Nations | Environment, Human Development |

Source: [78].

As can be seen in Table 2, the different indicators have different scopes of application and levels of analysis. This makes it clear that common indicators are needed that can be systematically applied to the measurement of GI or eco-innovative actions. Therefore, this shows us that there are few robust scales to measure GI [84]. Specifically, validated scales including specific indicators for measuring GI actions. Song et al. [85] indicate that there is no uniformity of measurement in this area, no permanent statistics are available, and there is a need to overcome deficiencies in methodology by designing activity-specific indicators, with the additional problem that GI is not yet considered a separate official sector of innovation, which makes data collection particularly costly.

It should be noted that the European Union is starting over the years to finance specific lines to respond to the problems of measuring the activities undertaken in GI. For example, Kemp and Pearson [86] highlight various measurement methods, such as survey analysis, patent analysis, and digital and documentary source analysis. In addition, the OCDE studies the strengths and weaknesses of existing indicators and methodologies and provides guidelines for future improvements in measuring GI. Therefore, we can point out that although some indicators exist for the measurement of GI, there is a need to develop more valid scales that can be applied to different geographical areas (regional, national, or international) and levels of analysis (companies vs. public organizations).

*4.4. The Role of Technology Centers in the Green Innovation Processes*

Technology centers have innovation as a primary field of action. Accordingly, these centers could be a great support for GI and its expansion by developing GI projects [1]. Technology centers can create projects where activities are developed to improve efficiency and results and to reduce environmental damage. In other words, they can implement innovative and sustainable projects. Hence, technology centers can implement actions to promote new technologies and eco-design [87]. These actions may include leadership actions and active participation of mixed consortiums with industries, European Union projects and other international initiatives, investments in the design of public experimental facilities, and technological collaboration.

On the other hand, according to Chen et al. [5], technology centers can support GI management by making free tools for measuring GI initiatives available to companies and governing bodies or by promoting the use of shared resources, free user-friendly tools, training support and specific tools in GI management. Moreover, other authors suggest that technology centers can implement and disseminate GI through outreach actions that promote eco-efficiency, energy efficiency, rational use of resources and materials, innovation that stimulates eco-innovation, and sustainability initiatives [88]. Provision by these centers of elements, equipment, spaces, and common services, together with the access to specialized human resources that they can offer, may also help to promote

the conditions that facilitate GI actions. In accordance, technology centers may play a very important role in disseminating and implementing GI. However, many of them are not particularly focused on GI. In fact, some authors (e.g., [1]) have noted this trend by suggesting that these centers make greater efforts in promoting GI.

## 5. Discussion

This research has summarized the theoretical basis of the concept of GI. As we have noted, the topic of GI has become increasingly important not only in academia but also in practice, as today's consumers are increasingly demanding a balance between production and environmental sustainability from companies and institutions. Moreover, due to changes in environmental regulations, more and more companies are showing an interest in implementing GI as a business strategy. This interest in GI also stems from the recognition that consumers or customers better value those companies that strive to carry out efficient environmental management and treat the environment responsibly. This manifestation by society is forcing companies to make certain decisions such as reviewing the concepts of value and profitability in their business models and considering the balance between the two objectives of short-term profitability and long-term sustainability. The growing interest in GI also stems from scientific research. In recent years, there has been a growing body of research investigating the topic of GI. However, findings on GI are still at an early stage of maturity. In this study, we have summarized some of the fundamental bases of the concept by reviewing the most significant theoretical foundations.

As we have proposed, GI collects ideas from the concepts of innovation and sustainability. Thus, it can be stated that the topic of GI arises from both innovation and sustainability literature. Accordingly, in this study, we have underpinned the concepts of innovation and sustainability before explaining GI. The purpose of this study was to explore what GI is, understand the main roots of the concept, and review the current state of the literature on the subject. We have also discussed the role that governing bodies and technology centers are playing in promoting GI. As indicated, governments and public institutions have developed several programs trying to promote GI. Moreover, many public policies have created indicators, control standards, and action plans to measure whether a company is truly eco-innovative or not. We have provided a compilation of the most-used indicators and how they are shaping the policies not only of the companies but also of governing bodies themselves. These policies, therefore, become a series of guidelines to follow to be an eco-innovative company. However, a real commitment to GI requires greater involvement of public institutions and governments.

As we note, companies can find a source of competitive advantage in GI. Therefore, this form of innovation can not only be positive for environmental sustainability but can also result in economic benefits. The paradigm shift towards a more sustainable world requires a commitment to sustainability measurement criteria that can be used as a benchmark for detecting the progress of companies and countries. This would entail a change in the economic model to reward the contributions to the common good, and not only the level of production and sale of products and services, as a pattern of growth. This change towards a new, more sustainable economic model, based on sustainable management, also requires a firm commitment to new green technologies. This requires governments and companies to collaborate in the investment and development of this type of technology.

Economic, social, and environmental sustainability are the major challenges facing society today. However, economic inequality, poverty, hunger, job insecurity, the destruction of ecosystems, the increase in global temperature, and the loss of biodiversity do not unfold in isolation. On the contrary, they are conditions that feedback on each other. For example, the destruction of ecosystems and local livelihoods also leads to higher levels of poverty and social exclusion for local populations. Therefore, the joint analysis of these dimensions of sustainability is essential if the living standards of the population are to be improved. In this sense, although GI is fundamentally aimed at improving environmental sustainability, it would eventually lead to improvements in other areas, such as the economic and social

spheres. Furthermore, although the importance of innovation is emphasized today, if it is not accompanied by the principles of sustainability, it may end up being more of a problem than a solution to today's challenges. In addition, tackling climate change requires an immediate and forceful response that involves all social actors, from governments and private companies to non-profit organizations and citizens. Therefore, an effective response to this problem requires that all these parties become involved in the participation or development of actions linked to sustainability and GI.

In short, this theoretical review has sought to offer a response to the solution to the challenge of sustainability by highlighting the importance of GI, integrating its theoretical bases, and reviewing some of its fundamental considerations. We hope that this review will serve as encouragement to continue exploring and applying GI in all types of settings and that it will provide a comprehensive understanding of its value and functioning so that it can continue to be deployed.

## 6. Conclusions

This study is intended to serve as a lynchpin for the continuation of the GI debate. In this way, the theoretical framework developed here could encourage further exploration of this concept. Although GI is a concept that is key to ensuring environmental protection and the future of the planet, there is still a long way to go in this field. We hope that the conceptual foundations explored throughout this study can be a guide to better understanding the concept and awaken greater academic interest. Future research could be directed at further deciphering the background of GI. For example, human resource management could play a key role in this regard by promoting GI processes. Although some progress has been made in deciphering this role, there is still a long way to go. Moreover, to underpin the importance of GI, more studies should be conducted to empirically demonstrate its effects on variables such as company performance or competitiveness. Another relevant question is to find out the role of organizational culture on the generation of GI, including what types of specific behaviors among employees are necessary to encourage it. We hope that the ideas summarized in this study inspire further exploration of these and other issues that affect the concept of GI, and will ultimately lead to a more sustainable economy.

**Author Contributions:** Conceptualization, J.G., Z.B. and R.C.; investigation, J.G., Z.B. and R.C.; writing—original draft preparation, J.G., Z.B. and R.C.; writing—review and editing, J.G.; visualization, J.G.; supervision, J.G.; project administration, J.G.; funding acquisition, J.G. All authors have read and agreed to the published version of the manuscript.

**Funding:** This research was funded by Generalitat Valenciana (ref. GV/2021/088) and Ministry of Science and Innovation of Spain (ref. PID2020-116299GB-I00).

**Institutional Review Board Statement:** Not applicable.

**Informed Consent Statement:** Not applicable.

**Data Availability Statement:** Not applicable.

**Acknowledgments:** Zina Barghouti is grateful to the Universitat Jaume I for funding her predoctoral fellowship (FPI-UJI).

**Conflicts of Interest:** The authors declare no conflict of interest.

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
