# Peer review of "Understanding Green Innovation: A Conceptual Framework"

_sustainability, doi:10.3390/su14105787_

Round 1

Reviewer 1 Report

Dear Editor

Thank you for giving me the chance to review the manuscript titled “Understanding green innovation: A conceptual framework” Here are my suggestions

  1. General information on environmental degradation is needed in the first paragraph of the introduction
  2. The motivation, as well as the significance of this study, is not clearly stated. Kindly expand on it
  3. This section needs improvement “What is meant by sustainability?”
  4. The concept of green innovation is wide; however, in this study, it’s not discussed fully discussed
  5. The discussion section needs significant improvement

Author Response

  1. General information on environmental degradation is needed in the first paragraph of the introduction

In accordance with your suggestion, we have included information detailing the environmental concerns: “In recent years, fossil carbon dioxide emissions have reached record highs, global average sea levels have risen by 20 cm from 1900 to 2018, and heat waves, forest fires and air pollution have in-creased. In addition, concentrations of major greenhouse gases have continued to increase in the last few years and the global mean surface temperature for the period 2017-2021 is among the warmest ever recorded. On the other hand, the frequency and intensity of extreme weather and climate events have increased in all regions of the world.”

  1. The motivation, as well as the significance of this study, is not clearly stated. Kindly expand on it

We have tried to justify more clearly the significance of the study. To this end, we have included the following: “Despite the progress achieved in this field of research and the growing attention this subject has received in recent years, there are very few literature reviews on this concept. The conceptual review carried out in this study, linking this concept with its foundations, as well as shedding light on its forms of measurement and application in public institutions and technology centers, is a pioneering work.”

  1. This section needs improvement “What is meant by sustainability?”

In accordance with your suggestion, we have expanded the section.

  1. The concept of green innovation is wide; however, in this study, it’s not discussed fully discussed.

In accordance with your comment, we have expanded the discussion on the concept of green innovation.

  1. The discussion section needs significant improvement

We have expanded this section according to your suggestion. We hope that the added content will considerably improve this section.

Thank you very much for all your comments and suggestions. We sincerely believe that they have helped to significantly improve the quality of the paper.

Reviewer 2 Report

The study is an important contribution to the initiation research in this field. In this sense, it is an important study that should be disseminated.

The particularly important is to pay the attention role of enterprises in the development of the GI. It is a pity that the authors did not try to supplement the theoretical considerations with examples taken from real business.

Referring to the content, it should be noted that the basic perception of the environmental requirements cannot (should not) allow talking about the GI. Legal regulations should be seen only as the basic framework for the business operations of enterprises. In addition, requiring the consideration of business strategy. This seems to require a clear indication.

Moreover, the authors refer to the possibility of understanding innovation in more detail in point 2.2 and following. They themselves show the complexity of the issue.

It is a pity that the authors did not try to detail the factors determining the dimensions of sustainable development. Indicating what includes: economic, social and environmental sustainability, through their connection with factors and verification indicators (Table 2?).

Please consider extending the concept of Green Innovation to include eco-friendly using, up to eco-disposal. (In addition to eco-design and eco-production).

The study lacks a formal summary. The discussion is not exhaustive of its essence. It should be treated as an in-depth analysis of the problem.

Informally, summary tasks are fulfilled by the last paragraph of the discussion.

Author Response

The study is an important contribution to the initiation research in this field. In this sense, it is an important study that should be disseminated.

The particularly important is to pay the attention role of enterprises in the development of the GI. It is a pity that the authors did not try to supplement the theoretical considerations with examples taken from real business.

In accordance with your suggestion, we have included a paradigmatic case of an example of a company that applies IG. This is the case of the company Patagonia Inc. whose business model focuses on manufacturing products following a design process that mitigates damage to the environment and that seeks to ensure that its economic activity contributes positively to the health of the planet:

“A paradigmatic case of a company that applies the fundamentals of GI is Patagonia Inc. This company manufactures sportswear following a management philosophy based on reducing negative environmental impact by creating recyclable, reusable and durable products. This requires that in the design of their products they apply the GI, so that they can achieve a product that is environmentally sustainable. This would be a good example of a company that follows the precepts of what is meant by environmental management and has been commonly held up as a model of companies that apply eco-innovation (Guinot, 2021).”

Referring to the content, it should be noted that the basic perception of the environmental requirements cannot (should not) allow talking about the GI. Legal regulations should be seen only as the basic framework for the business operations of enterprises. In addition, requiring the consideration of business strategy. This seems to require a clear indication.

We agree with their assessment that legal regulations are not the only reason for applying GI. In fact, we indicate on page 7 that “given that in recent years consumer awareness of sustainability is growing, many companies have used the green agenda as an engine for achieving a business model based on GI [74]. Thus, GI practices are a logical response to the requirements of the customer, who is willing to pay more for sustainable products and expect increasingly re-sponsible products and services from firms [1,75]. In fact, previous research (e.g., [6,76]) indicates that GI initiatives contribute to the improvement of the company's performance and competitiveness, since GI practices increase product value and create competitive advantages by means of ecological differentiation.”

Moreover, the authors refer to the possibility of understanding innovation in more detail in point 2.2 and following. They themselves show the complexity of the issue.

It is a pity that the authors did not try to detail the factors determining the dimensions of sustainable development. Indicating what includes: economic, social and environmental sustainability, through their connection with factors and verification indicators (Table 2?).

Thank you for your note. On page 5 we explain what each of these dimensions consists of. In particular, we note that “Economic sustainability refers to the use of practices and activities that are economically profitable, but also socially and environmentally responsible [45,46]. This means seeking growth and economic profitability without forgetting social welfare and environmental care. Moreover, environmental sustainability refers to the use of natural resources in an efficient and rational way to improve current social welfare without compromising the quality of life of future generations [47,43]. Finally, social sustainability refers to the achievement of a balance between the two dimensions mentioned above without generating poverty, exclusion, and inequality [48]”. The indicators in Table 2 are based on these different dimensions in order to measure IG in different contexts.

Please consider extending the concept of Green Innovation to include eco-friendly using, up to eco-disposal. (In addition to eco-design and eco-production).

In line with your suggestion, we have extended the review of Green Innovation concept by also referring to the processes you mention:Along the same lines, some authors (e.g. Jabbour et al., 2005; Albort-Morant et al., 2016) define green innovation as the development of sustainable products and processes through the use or adoption of environmentally friendly raw materials during the manufacturing or design process. This process also involves the application of the principle of eco-design or eco-production. That is to say, the environment is taken into consideration from the moment a product or process idea is conceived, and not only at the end of its useful life (MacDonald, 2015)”. 

“Furthermore, the use of environmentally friendly raw materials, or what is known as "eco-friendly using", which goes hand in hand with GI, leads to several relevant benefits that can differentiate the company from competitors (Hwang et al., 2019). These benefits include cost reduction, improved company reputation, increased business opportunity and attracting new investors and customers (Hwang et al., 2019; Tang et al., 2020). Therefore, GI is associated with better environmental management and environmental performance. In this way, the GI in processes and products not only reduces the negative environmental impact of the business, but also increase organisational performance through waste and cost reduction (Weng et al., 2015).”

The study lacks a formal summary. The discussion is not exhaustive of its essence. It should be treated as an in-depth analysis of the problem.

In accordance with your comment, we have completed the discussion section, so that a more detailed analysis is provided. We have also included a final section with conclusions.

Informally, summary tasks are fulfilled by the last paragraph of the discussion.

Thank you very much for all your comments and suggestions. We sincerely believe that they have helped to significantly improve the quality of the paper.

Reviewer 3 Report

The authors aim to address the concept of green innovation from a theoretical perspective. The authors aim to integrate the existing literature and provide a theoretical framework to serve as a reference to give continuity to future research on the subject. The article does archive this aim. The article, however, needs some further improvements in order to be aligned with the aims, scope and standards of the journal: Sustainability. I have listed some specific comments that might help the authors further enhance the manuscript's quality.

  1. Abstract: the abstract needs revisions

The abstract should be a total of about 200 words maximum. The abstract should be a single paragraph and should follow the style of structured abstracts, but without headings: 1) Background: Place the question addressed in a broad context and highlight the purpose of the study; 2) Methods: Describe briefly the main methods or treatments applied. Include any relevant preregistration numbers, and species and strains of any animals used. 3) Results: Summarize the article's main findings; and 4) Conclusion: Indicate the main conclusions or interpretations. The abstract should be an objective representation of the article: it must not contain results which are not presented and substantiated in the main text and should not exaggerate the main conclusions.

  1. Research originality.

Please clarify how is your article adds new knowledge to the body of knowledge that already exists in a research area.

  1. Methodology

The authors need to offer a concrete research framework. A diagram might be helpful.

  1. Discussion

This section should focus on explaining and evaluating what the authors have found, demonstrating how it relates to your literature review and research questions. You may also make an argument in support of your overall arguments.

  1. Conclusion

The authors need to draft a conclusion section. The conclusion of the article should be clear and concise and try to address the question and explain how you have met the research objective raised in the introduction.

Author Response

The authors aim to address the concept of green innovation from a theoretical perspective. The authors aim to integrate the existing literature and provide a theoretical framework to serve as a reference to give continuity to future research on the subject. The article does archive this aim. The article, however, needs some further improvements in order to be aligned with the aims, scope and standards of the journal: Sustainability. I have listed some specific comments that might help the authors further enhance the manuscript's quality.

  1. Abstract: the abstract needs revisions

The abstract should be a total of about 200 words maximum. The abstract should be a single paragraph and should follow the style of structured abstracts, but without headings: 1) Background: Place the question addressed in a broad context and highlight the purpose of the study; 2) Methods: Describe briefly the main methods or treatments applied. Include any relevant preregistration numbers, and species and strains of any animals used. 3) Results: Summarize the article's main findings; and 4) Conclusion: Indicate the main conclusions or interpretations. The abstract should be an objective representation of the article: it must not contain results which are not presented and substantiated in the main text and should not exaggerate the main conclusions.

In accordance with their recommendations, we have rewritten the abstract of the paper.

  1. Research originality.

Please clarify how is your article adds new knowledge to the body of knowledge that already exists in a research area.

As you recommend, we have more clearly stated in the introduction of the article its contribution and academic interest. In particular, we have added the following statement:

“Despite the progress achieved in this field of research and the growing attention this subject has received in recent years, there are very few literature reviews on this concept. The conceptual review carried out in this study, linking this concept with its foundations, as well as shedding light on its forms of measurement and application in public institutions and technology centers, is a pioneering work.”

  1. Methodology

The authors need to offer a concrete research framework. A diagram might be helpful.

This is a theoretical article that does not follow a particular research methodology beyond the reading and review of the existing literature on the topic addressed. Therefore, it does not include a section on the methodological or research framework used.

  1. Discussion

This section should focus on explaining and evaluating what the authors have found, demonstrating how it relates to your literature review and research questions. You may also make an argument in support of your overall arguments.

In accordance with your indications, we have completed this section so that the actual analysis is now more complete.

  1. Conclusion

The authors need to draft a conclusion section. The conclusion of the article should be clear and concise and try to address the question and explain how you have met the research objective raised in the introduction.

In accordance with your suggestion, we have included a final section with the conclusion.

Thank you very much for all your comments and suggestions. We sincerely believe that they have helped to significantly improve the quality of the paper.

Reviewer 4 Report

The article covers an emerging topic in management related to the intersection of sustainability and innovation on the agenda of companies.

The introduction is clear. Nevertheless, it states some objectives that are not fulfilled by the rest of the article. Eg: "Thus, to consolidate a theoretical framework which provides continuity and consistency to the findings, this study offers a complete review of the concept of GI."

Chapter 2 and 3 provide a description of innovation and sustainability that is well known and documented. 

Chapter 4 presents a set of approaches to the concept of GI, that lack coherence among them and with the rest of the article. It starts by developing the concept of green innovation (supported by figure 2 that is structurally very similar to figure 1) and dissertates on the "role of administration promoting green innovation" using mostly European Union related initiatives. The measurement part presents a set of potential indicators that have little coherence with the previously presented concepts. It finally ends approaching the role of technology centers. This chapter should be revised in order to have a coherent approach.

The discussion in chapter 5 is weakened by the same problem of lack of coherence although it approaches and discusses correctly the topics in chapter 4. (It should be noticed that GI and IG are used interchangeably ... it should be corrected)

In general, it is well written article that, although it claims to be a theoretical approach to the topic, does not bring new models to literature and lacks structure, approaching diverse items with no "conducting thread" among them.

As a final note, most of the references are over 5 years old.

Author Response

The article covers an emerging topic in management related to the intersection of sustainability and innovation on the agenda of companies.

The introduction is clear. Nevertheless, it states some objectives that are not fulfilled by the rest of the article. Eg: "Thus, to consolidate a theoretical framework which provides continuity and consistency to the findings, this study offers a complete review of the concept of GI."

As you indicate, we have corrected the introduction section to be more precise with the definition of objectives in accordance with the contribution made.

Chapter 2 and 3 provide a description of innovation and sustainability that is well known and documented.

Chapter 4 presents a set of approaches to the concept of GI, that lack coherence among them and with the rest of the article. It starts by developing the concept of green innovation (supported by figure 2 that is structurally very similar to figure 1) and dissertates on the "role of administration promoting green innovation" using mostly European Union related initiatives. The measurement part presents a set of potential indicators that have little coherence with the previously presented concepts. It finally ends approaching the role of technology centers. This chapter should be revised in order to have a coherent approach.

As explained at the end of Chapter 3, sustainability or sustainable business models are based on GI as an essential tool. The development of GI is an essential issue to be able to implement sustainability, hence they are two intrinsically related concepts. Specifically, we note that: "...companies that share the importance of sustainability as one of their core values have GI as a primary tool. Hence, GI has become a common strategy for companies oriented towards the common good and concerned with sustainability issues". This is also noted at the beginning of section 4.1 when we state that: "GI has become one of the most important strategic tools for effective sustainable de-velopment [73]. In the past, investing in environmental activities was unnecessary, however, strict environmental regulations and popular environmentalism have changed the competitive rules in practice [1]. This has led to an expansion of methods and processes linked to environmental sustainability, such as GI”. The intrinsic relationship between sustainability and environmental management is also contemplated in the two figures we provide, whose similarity reflects the direct connection between these two concepts or variables.

On the other hand, we believe that the structure of Chapter 4 is coherent, first explaining the concept of GI and then addressing in depth some issues linked to the concept of GI, such as the role of the administration, forms of measurement and the role of technology centers. In any case, to underpin this chapter we have expanded the explanation in the first section, so that the concept of GI is more accurately stated and its relationship with the other sections can be better understood.

The discussion in chapter 5 is weakened by the same problem of lack of coherence although it approaches and discusses correctly the topics in chapter 4. (It should be noticed that GI and IG are used interchangeably ... it should be corrected)

We have corrected these typos and improved the discussion.

In general, it is well written article that, although it claims to be a theoretical approach to the topic, does not bring new models to literature and lacks structure, approaching diverse items with no "conducting thread" among them.

As a final note, most of the references are over 5 years old.

Thank you very much for all your comments and suggestions. We sincerely believe that they have helped to significantly improve the quality of the paper.

Round 2

Reviewer 3 Report

The authors have revised the article according to my suggestions.